# Effects of Dietary Koumine on Growth Performance, Intestinal Morphology, Microbiota, and Intestinal Transcriptional Responses of *Cyprinus carpio*

**DOI:** 10.3390/ijms231911860

**Published:** 2022-10-06

**Authors:** Qiujie Wang, Dongjie Wang, Zhiheng Zuo, Bin Ye, Zaijie Dong, Jixing Zou

**Affiliations:** 1College of Marine Sciences, South China Agricultural University, Guangzhou 510642, China; 2Key Laboratory of Freshwater Fisheries and Germplasm Resources Utilization, Ministry of Agriculture and Rural Affairs, Freshwater Fisheries Research Center, Chinese Academy of Fishery Sciences, Wuxi 214081, China

**Keywords:** herbal extract, biochemical index, cellular regulation, proliferation, apoptosis

## Abstract

*Gelsemium elegans* *Benth*. (GEB) is a traditional medicinal plant in China, and acts as a growth promoter in pigs and goats. Koumine (KM) is the most abundant alkaloid in GEB and produces analgesic, anti-cancer, and immunomodulatory effects. KM can be used as an aquatic immune stimulant, but its growth-promoting effects and transcriptional mechanisms have not been investigated. Diets containing KM at 0, 0.2, 2, and 20 mg/kg were fed to *Cyprinus carpio* for 71 days to investigate its effects on growth performance, intestinal morphology, microflora, biochemical indicators, and transcriptional mechanisms. *Cyprinus carpio* fed with KM as the growth promoter, and the number of intestinal crypts and intestinal microbial populations were influenced by KM concentration. KM increased the abundance of colonies of *Afipia*, *Phyllobacterium*, *Mesorhizobium*, and *Labrys*, which were associated with compound decomposition and proliferation, and decreased the abundance of colonies of pathogenic bacteria *Methylobacterium-Methylorubrum*. A total of 376 differentially-expressed genes (DEGs) among the four experimental groups were enriched for transforming growth factor-β1 and small mother against decapentaplegic (TGF-β1/Smad), mitogen-activated protein kinase (MAPK), and janus kinases and signal transducers and activators of transcription (Jak/Stat) signaling pathways. In particular, *tgfbr1*, *acvr1l*, *rreb-1*, *stat5b*, *smad4*, *cbp*, and *c-fos* were up-regulated and positively correlated with KM dose. KM had a growth-promoting effect that was related to cell proliferation driven by the TGF-β1/Smad, MAPK, and Jak/Stat signaling pathways. KM at 0.2 mg/kg optimized the growth performance of *C. carpio*, while higher concentrations of KM (2 and 20 mg/kg) may induce apoptosis without significantly damaging the fish intestinal structure. Therefore, KM at low concentration has great potential for development as an aquatic growth promotion additive.

## 1. Introduction

Aquaculture provides an important source of protein for humans. However, the massive use of antibiotics and chemical drugs in aquaculture to reduce the occurrence of diseases has caused serious consequences, namely the development of drug-resistant pathogens and pollution of the environment [1]. Healthy, environmentally friendly, and efficient methods for fish cultivation have essential economic benefits. Herbs can act as environmentally friendly feed additives to prevent disease and promote growth performance in farmed fish [1,2,3].

*Gelsemium elegans Benth*. (GEB), native to Southeast Asia and China, is a readily available and inexpensive herbal medicine that has been used in traditional Chinese medicine to treat skin ulcers and dermatitis, and as a treatment for cancer-related pain [4,5]. GEB has been tested in livestock production for its effects linked to mitogen-activated protein kinase (MAPK) and PI3 kinase/mTOR signaling (PPRK) in pigs which has been recommended as an antibiotic alternative [6]. Interestingly, GEB has been successful in promoting digestion and relieving pain in pigs, goats, and cattle and these effects have been listed in the Chinese Veterinary Pharmacopeia [7]. Moreover, the combination of GEB with ginseng increased the daily weight gain of pigs by 16.6% and the feed conversion ratio by 18.2%, accelerating the growth of pigs [6]. The affordability and availability of GEB, as well as its ease of decomposition, make it an environmentally friendly feed additive [8]. Therefore, it is necessary to study the active ingredients and effects of GEB in depth.

The main active constituents of GEB are alkaloids, among which koumine (KM) is the most abundant. KM is a lipophilic molecule with poor solubility in water (<1 mg/mL). It possesses analgesic, anti-cancer, neuroprotective, immunomodulatory, and anti-inflammatory effects [9,10,11]. KM demonstrated a a dose-dependent inhibition of *Tetrahymena thermophila* proliferation, accompanied by the significant (*p* < 0.05) elevation of antioxidant enzyme activity and apoptosis, but induced DNA damage at high concentrations in one study [12]. Overall, KM has rapid pharmacokinetic characteristics and concentration dependence compared to other alkaloids [13]. Studies on the effectiveness of KM for animals, especially poultry and livestock, have focused on its growth efficacy and pharmacokinetics. However, the research on KM in aquaculture is not extensive [10,14,15,16].

Growth and development are intimately connected with signaling between cells and tissues [17]. Cellular signaling pathways such as transforming growth factor β1 (TGF-β1) and MAPK pathways are particularly essential for interactions between an organism and its environment during growth and development [18]. The transforming growth factor-β1 and small mother against decapentaplegic (TGF-β1/Smad) pathway is a regulator of type I collagen expression. It can affect muscle stiffness, and is a mediator of cell growth and differentiation [19,20,21,22]. The MAPK family of kinases include p38 MAPK, Jun N-terminal kinase (JNK), and extracellular signal-regulated kinase (ERK). MAPK studies in fish have focused on hormone induction, differentiation, and proliferation [23,24,25]. Janus kinases and signal transducers and activators of transcription (Jak/Stat), MAPK, and TGF-β signaling can regulate stem cell pluripotent, which are attached to development and proliferation [26]. Interestingly, α-helical antimicrobial peptides (AMP) was applied to enhance fish intestinal immunity, whose addition to cultures improved the growth performance of grass carp *Ctenopharyngodon idella* via Jak/Stat signaling pathway [27]. These studies link cellular pathways to biological growth as well as the intestinal microbiota, which are critical for growth and developmental processes in fish [28].

Until now, there has been no study on the effects of KM on fish growth performance. Therefore, we sought to determine the effects of KM on growth performance, histopathology, intestinal microbiota, and biochemical indexes in common carp (*Cyprinus carpio* L.). To link its effects to the TGF-β1/Smad, MAPK, and Jak/Stat signaling pathways, we monitored these effects using intestinal transcriptome analysis. Our study provides the data to support the application of KM as an aquatic growth additive.

## 2. Results

### 2.1. Growth Performance and Intestine Histological Analysis

The data collected in this paper at 71 days rather than eight weeks, which is common in culture experiments, is because there were no significant differences in fish growth data during the first 45 days of the culture process. No fish mortality or visible abnormalities were observed during the trial. We compared the effects of KM on the growth performance of *C. carpio* through measurements of weight, body length, condition factor (CF), weight gain (WG), viscerosomatic index (VSI), and hepatosomatic index (HSI). We treated KM levels logarithmically and performed a polynomial regression analysis in weight, CF, and WG to estimate whether the effect of KM on the growth performance of *C. carpio* was significant (Appendix A). The 0.2, 2, and 20 mg/kg treatment groups displayed increases in body weight, length, WG, and a significant increase in CF. The body weight of fish was significantly increased in the 0.2 and 2 mg/kg experimental groups. In addition, we found significant (*p* < 0.05) increases in WG (5.37 ± 2.31%), VSI (0.09 ± 0.02%), and HSI (0.06 ± 0.02%) for the 0.2 mg/kg group (Table 1).

The external morphological structure of the *C. carpio* intestine remained intact in all groups, although KM levels in the diet correlated with evidence of cellular hyperplasia. Compared to the control, dense crypt structures were observed in the intestine of *C. carpio* for all treatment groups. We also found obvious signs of epithelial, phagocytes, and submucosal cell proliferation in the experimental groups. However, the 20 mg/kg group showed disturbed cell arrangements as well as epithelial cell vacuolation (Figure 1).

### 2.2. Effect of KM on the Intestinal Microflora of C. carpio

To explore the effect of KM on the intestinal microbial community of *C. carpio*, we preprocessed the data to remove low-quality and ambiguous sequences to yield a total of 428,757 qualified 16S rRNA sequences. The numbers of valid sequences generated per sample ranged from 30,084 to 56,546 and the average length of valid sequences was 375 bp. This enabled the definition of 2735 operational taxonomic units (OTUs) at the 97% similarity level (Appendix A). The OTU number, based on a rarefaction analysis indicated that sufficient sampling depth was achieved for each sample and that our results were valid.

The Chao1 richness index of alpha diversity was lower in the 0.2 and 20 mg/kg groups compared with the control (Appendix A) as was the Ace richness index (Appendix A). In contrast, the Shannon and Simpson indices showed an increasing trend that was KM dose-dependent (Appendix A). The Good’s coverage index for three treatments was significantly (*p* < 0.05) higher than the control and almost reached the level of >99% (Appendix A). It suggested that the identified sequences represented the majority of bacteria present in the intestines of our fish.

The effects of dietary KM levels on the intestinal microbiota were assessed by bacterial communities. At the phylum level, the Proteobacteria and Bacteroidota dominated in all groups. However, the Fusobacterial phylum differed corresponding to 0.0031%, 0.0032%, 0.0639%, and 0.0193% for the control and test groups, respectively (Figure 2a). At the genus level, *Bradyrhizobium*, *Methylocyclus*, and *Vibrio* were the most representative regardless of treatment (Figure 2b), and at the species level were represented by five dominant genera: *Afipia*, *Phyllobacterium*, *Mesorhizobium*, *Labrys* and *Methylobacte-rium-Methylorubrum*. The abundance of *Afipia* in the 0.2 mg/kg group and *Phyllobacterium* in the 2 mg/kg group increased significantly. However, the abundance of *Mesorhizobium* and *Labrys* in 20 mg/kg group visibly decreased. Furthermore, the three test groups displayed significant declines in *Methylobacterium-Methylorubrum* according to dose (Figure 2c).

### 2.3. Intestine Antioxidant Indicators

As shown in Figure 3, there were antioxidant indices for the effects of KM on *C. carpio* intestines in this study. Compared with the control group, the Superoxide dismutase (SOD) activity (Figure 3a) and malondialdehyde (MDA) content (Figure 3b) of each experimental group (0.2, 2, and 20 mg/kg) were significantly increased. The catalase (CAT) activity (Figure 3c) of 0.2 mg/kg group was significantly decreased. The glutathione (GSH) level of the 20 mg/kg group was significantly lower than that of the control group (Figure 3e). The acid phosphatase (ACP) activities were lower in the 0.2, 2, and 20 mg/kg groups than in the control group, and the reduction was particularly significant in the 0.2 mg/kg and 20 mg/kg groups (Figure 3g). 

### 2.4. Transcriptome Sequencing and Verification

We conducted transcriptome analyses of triplicate samples of each group and 94.85 GB of clean data were obtained and reached >6.94 GB per sample with errors < 0.025%. The clean reads that we obtained ranged from 47,470,704 to 62,952,880 and the average percentage of Q20 and Q30 scores and GC content were 98.33, 94.87, and 46.68%, respectively (Appendix A). These data indicated that the RNA-seq procedure produced high-confidence sequences. After the removal of redundant sequences, 55,661 genes including 49,578 known and 6083 new genes were detected as expressed in the intestine (Appendix A). The total number of gene sequences that we obtained ranged from 21,068 to 24,618 per sample (Appendix A). These included 18,435 co-expressed genes and PCA indicated a high correlation between samples (Appendix A).

The differential expression analysis identified 7712 differentially-expressed genes (DEGs) across all groups and included 376 co-differentially expressed genes among all groups (Figure 4a). Furthermore, 3017 DEGs (1454 up-regulated and 1563 down-regulated) were markedly enriched in 0 vs. 0.2 mg/kg group, and 4256 DEGs (2057 up-regulated and 2199 down-regulated) were markedly enriched in 0 vs. 2 mg/kg group. Moreover, 3015 DEGs (1446 up-regulated and 1569 down-regulated) were markedly enriched in the 0 vs. 20 mg/kg group. In particular, the 0 mg/kg vs. 2 mg/kg group contained more DEGs, and the 2 mg/kg group displayed more up-regulated DEGs (Figure 4b, Appendix A).

To verify the accuracy of the transcript sequencing data, random genes were selected from the transcriptome for evaluation of their expression profiles in intestines using qRT-PCR. We compared fold change in expression levels between RNA-seq and qRT-PCR corresponding to the 2 and 20 mg/kg groups (Figure 4d,e) and the correlation indices were 0.85 and 0.9, respectively (Figure 4f,g). The expression patterns of these genes were similar between the qRT-PCR and RNA-seq, indicating that our RNA-seq data were reliable. 

### 2.5. Identification and Functional Annotation of DEGs

To further study the effect of KM on the intestinal function of *C. carpio*, we carried out GO and Kyoto Encyclopedia of Genes and Genomes (KEGG) annotation analyses of the 376 co-expressed DEGs (Appendix A). The top term for molecular function was binding and for the biological process, the top term was cellular process, indicating that ion signal transduction with the involvement of cellular receptors plays a significant role in the response of *C. carpio* to KM in their diets (Figure 5a,b). Furthermore, the KEGG analysis indicated that signal transduction in the environmental information processing and cell growth and death in cellular processes were also markedly enriched (Figure 5c). This demonstrated that KM invoked cell growth and death responses in *C. carpio*. We combined the KEGG pathway analysis of the DEGs from the 0.2, 2, and 20 mg/kg groups (Figure 5e–g). The DEGs were co-enriched in MAPK and TGF-β signaling pathways (Figure 5d). These data linked KM feeding with proliferative responses.

### 2.6. Expression of Genes Related to Cell Proliferation

To investigate the effect of KM on the transcriptional machinery of the *C. carpio* intestine, key genes were selected for quantification. The expressed target genes related to MAPK and TGF-β signaling (transforming growth factor beta receptor type 1, *tgfbr1*; activin receptor type 1, *acvr1l*; Ras response element binding protein 1, *rreb-1*; signal transducer and activator of transcription 5B, *stat5b*; Smad family member 4, *smad4*; CREB-binding protein, *cbp*, and c-Fos induced growth factor, c-*fos*) were related to cell proliferation. The relative expression of these genes in each experimental group (0.2, 2, and 20 mg/kg) showed a trend of down-regulation (0.2 mg/kg) and then up-regulation (2 and 20 mg/kg) according to dose (Figure 6a). We subjected the relative expression of genes to polynomial fit analysis, and found that the relative expression of *c-fos* in the 2 and 20 mg/kg groups showed significant up-regulation, indicating that the cellular process pathways for these two groups were quite active (Figure 6b).

## 3. Discussion

### 3.1. KM Improvement of Growth Performance

In studies conducted in China, GEB research has focused on its structural composition, pharmacological effects and its growth-promoting effects on poultry and livestock [4,7,29]. In aquaculture, the growth-promoting effects of GEB on aquatic animals have not been systematically studied, and the current work intends to fill this knowledge gap using the guidance of previous studies [1,12,30]. As a major component of GEB, KM improved the growth and intestinal health of *C. carpio.* In particular, body weights and length increased significantly compared to the control. WG significantly increased in the 0.2 mg/kg group, indicating that 0.2 mg/kg KM was the most effective concentration in our study. 

Morphological observations of the *C. carpio* intestines indicated significant signs of cellular proliferation of epithelial, goblet, and submucosal cells, as well as increases in crypt depths. Intestinal villi and microvilli provide a large surface area for efficient metabolism and nutrient absorption. Intestinal stem cells continuously undergo self-renewal and give rise to the transit of cells that move up the crypt villi axis that form various differentiated cell types to maintain intestinal epithelial integrity [31]. Further, growth performance depends on the digestive and absorptive capacity of fish, especially in metabolism and nutrient absorption [32]. For example, dietary dandelion improved intestinal morphology and immunity by increasing the number of intestinal cupped cells and decreasing the number of epithelial lymphocytes in the pompano fish *Trachinotus ovatus* [33]. Ghrelin addition to the diet of juvenile grass carp promoted intestinal cell proliferation and increased protein absorption and metabolism [3].

In our study, KM experimental groups exhibited enhanced cell proliferation and crypt hyperplasia, which could be indirectly related to KM dose. The 0.2 mg/kg group showed the best growth performance, but the intestinal villous epithelial cells did not proliferate as much as the 2 and 20 mg/kg groups, suggesting that KM at 2 and 20 mg/kg may accelerate the cell renewal process but not for effective nutrient absorption. A study found that 2 mg/kg of KM induced apoptosis in mouse colorectal cancer cells (CRCs), suggesting that high concentrations of KM can induce other cellular processes [34]. Our transcriptome analysis revealed the presence of genetic changes related to cell proliferation and apoptosis (such as *tgfbr1*, *acvr1l*, *stat5b*, and *smad4*), and we do not exclude the possibility that high concentrations of KM induce apoptosis in intestinal epithelial cells of *C. carpio*.

### 3.2. KM Enhancement of C. carpio Intestinal Flora

The intestine is the place of nutrient absorption in fish. Intestinal health and growth performance are directly associated with the balance of intestinal flora [35]. Regulating fish intestinal flora and its activity through dietary additives can improve fish health [33]. In the present study, the Proteobacteria and Bacteroidota comprised the core microbiota of the fish intestine at the phylum levels, consistent with previous studies [36]. It is related to water quality, population density, diet, and environment [37]. We found three primary genera in our samples (*Bradyrhizobium*, *Methylovirgula*, and *Vibrionimonas*). The abundance of *Phyllobacterium*, *Mesorhizobium*, and *Labrys* elevated was found in the 0.2 mg/kg group and *Afipia*, *Mesorhizobium*, and *Labrys* were abundant in the 2 mg/kg group. Meanwhile, the abundance of the pathogen *Methylobacterium-Methylorubrum* significantly decreased in the KM experimental group. These results suggested that dietary KM can adjust the ecological niches of the intestinal microflora in *C. carpio*. The presence of *Afipia* may be associated with the degradation of water pollutants [38]. *Phyllobacterium* are a potential biomarker for hepatocellular carcinoma and are related to cellular proliferation [39]. The potential beneficial effects of *Mesorhizobium* on plants are mineral nutrient solubilization, abiotic stress tolerance, and nitrogen fixation, but its effects on the *C. carpio* intestine are currently unknown [40]. Members of the genus *Labrys* are capable of degrading and utilizing fluorobenzene as a sole carbon source [41]. *Methylobacterium-Methylorubrum* is a common pathogenic bacterium capable of infecting immune-compromised fish [42]. Our results suggested that the 0.2 mg/kg KM increased the abundance of microorganisms associated with contaminant degradation and mineral nutrient absorption, which may favor nutrient metabolism in the intestine. However, the abundance of *Afipia* decreased in 2 and 20 mg/kg KM groups, while *Phyllobacterium* increased, reflecting the finding that KM at this concentration may promote intestinal cell proliferation and accelerate cellular processes; however, it is detrimental to the catabolic absorption of substances.

### 3.3. KM Protected C. carpio Intestinal Cells from Oxidative Damage

SOD, CAT, GSH, and LDH are some major antioxidant defense systems in cells [43,44]. It is well-known that SOD and CAT work together to reduce oxidative damage. T-AOC is an indicator of lipid peroxidation sensitivity and can be used to reflect the health and antioxidant capacity of tissues [45,46,47,48]. In the study, the most significant changes in the three experimental (0.2, 2, and 20 mg/kg) groups were the simultaneous decrease in SOD activity and MDA content, and a significant decrease in CAT activity in the 0.2 mg/kg group. Moreover, the levels of GSH in the 20 mg/kg KM group significantly declined. A possible reason for these results may be that KM preferentially utilizes reactive oxygen species (ROS) compared to antioxidant enzymes without changing the total antioxidant capacity of carp, thereby reducing the activity of antioxidant enzymes [49,50,51]. Although antioxidant enzyme activity decreased, there was no oxidative damage, which indicated that KM induces cells to use ROS for apoptosis. A report showed that when KM reached a concentration of 400 ug/mL, ROS was used to promote phosphorylation of ERK and others, leading to apoptosis [52]. Moreover, a study demonstrated that KM significantly reduced ROS and could effectively protect cells from H_2_O_2_^−^ induced damage, and our results are consistent [53]. AKP is a multifunctional enzyme involved in the basic functions of the organism, while ACP is a typical lysosomal enzyme that plays a role in killing and digesting pathogens in the immune response. ACP and AKP activity significantly increase with pathogen infection [43,54]. Our results are similar to those of two studies, where the decrease in ACP and AKP activity in grass carp and crucian carp confirmed the absence of exogenous oxidative stress production in fish [54,55]. As mentioned, KM can induce cells to consume ROS and avoid oxidative stress. Variations in KM concentration appeared to activate different cellular processes, promoting cell proliferation at 0.2 mg/kg.

### 3.4. Analysis of Cell Proliferation Process by KM

Transcriptome analysis revealed the presence of 376 DEGs in the *C. carpio* intestine, which were enriched in cellular processes, signal transduction in cell growth, and death. These included TGF-β1/Smad, MAPK, and Jak/Stat signaling (Figure 7). Binding of TGF-β1 to TGFβRI and TGFβRII is the first step in the TGF β/Smad signaling to activate the transcriptional network and thereby drive the phenotypic transformation of cell activation. TGF-β/Smad signaling can be activated by retinoid X receptor α which promotes cell proliferation [56]. We quantified genes involved in the TGF-β/Smad signaling pathway and found that the relative expression of *tgfbr1, acvr1l*, and *smad4* was up-regulated in the 2 and 20 mg/kg groups, but down-regulated in the 0.2 mg/kg group. Our experimental results suggested that KM promoted the growth of *C. carpio* most likely due to proliferation regulated by TGF-β [57]. The expression levels of these genes were positively related to KM dose, implying an acute effect of KM on the regulation of these genes. The MAPK pathway regulates physiological processes such as cell proliferation and RAS proteins are central regulators of growth factor-induced cell proliferation and survival and upstream factors of ERK signaling and proliferation [24,58]. The c-Fos and c-Jun protooncogenes are stimulated by extracellular ERK, which results in proliferation, differentiation, apoptosis, and transcription [59]. In addition, the *rrbe-1* gene is a transcriptional effector linked to the epithelial–mesenchymal transition, and the *cbp* gene is a key factor for the establishment and activation of enhancer-mediated transcription [60]. We found that the relative expression of *rrbe-1*, *cbp*, and *c-fos* genes were up-regulated in a dose-dependent manner. Furthermore, the relative expression of the *stat5b* gene was gradually up-regulated with KM dose, which acted as a negative regulator to suppress excessive cellular transcription. Overall, our results indicated that the primary effect of KM was through TGF-β1/Smad, MAPK, and Jak/Stat signaling pathways, and the expression of these genes was positively correlated with KM dose. In particular, *c-fos* was induced >30-fold, indicating that cellular processes were ongoing.

Previous studies have revealed that GEB significantly promoted the growth performance of *Megalobrama amblycephala*, consistent with our results using KM [1]. These effects are most likely due to enhanced nutrient absorption capacity through cell proliferation in the intestine. Notably, all our experimental KM groups had growth-promoting effects on *C. carpio*, although there were still differences between different KM concentration groups. In the 0.2 mg/kg KM group, the growth performance was the best, while related pathway gene expression was slightly down-regulated compared with the control. In contrast, these genes were up-regulated in the 2 and 20 kg/mg KM groups without optimal growth performance. A possible reason is that these doses have activated another cellular process, apoptosis. KM at concentrations of 50–400 µg/mL was found to have a protective effect against apoptosis induced by H_2_O_2_ and lipopolysaccharide, which may be beneficial to cell proliferation [53,61]. KM is associated with apoptotic effects in animals, which is strongly related to the concentration of KM [62,63]. We suggest that KM induces both proliferation and apoptosis occurrence in *C. carpio* intestinal cells, and a concentration threshold exists between the conversion of the two processes, which exists between 0.2 and 2 mg/kg. However, the threshold needs to be confirmed by further studies. Therefore, the presence of a low concentration of KM can induce the proliferation of *C. carpio* intestinal cells and the effective absorption of nutrients to achieve the best growth promotion effect. On the other hand, the increase in KM concentration induces apoptosis and accelerates cell renewal, which is detrimental to nutrient absorption.

## 4. Materials and Methods

All experiments were performed in accordance with the Guidelines for the Care and Use of Laboratory Animals in China and approved by the Institutional Animal Care and Use Committee (IACUC) of South China Agricultural University, China.

### 4.1. Experimental Diets

Basal diet composition and KM concentration settings were based on previous safety studies with 0, 0.2, 2, and 20 mg/kg KM added to experimental diets [30]. All basic ingredients were thoroughly mixed with KM, fish oil, and water and then formed into a sedimentary feed by a pelletizer. After forming, the pellets were air-dried to <10% moisture and all diets were sealed and stored at −20 °C until use. The KM used in this study was provided by Chendu Must Bio-technology Co., Ltd. (Chendu, China).

### 4.2. Experimental Fish and Culture Conditions

Fish experiments were performed in a recirculating aquaculture system in the laboratory of College of Marine Science, South China Agricultural University. The common carp strain FFRC were obtained from the Freshwater Fisheries Research Center, CAFS (Wuxi, China). All fish were fed with the basal diet for two weeks to adapt to the experimental diet and culture conditions (26.0–28.0 °C, pH 7.3–7.5, dissolved oxygen 5.5–6.1 mg/L). After acclimation, 360 fish of similar size (5.21 ± 0.05 g, 5.7 ± 0.1 cm) were randomly distributed into 12 tanks (60 × 40 × 35 cm) at 30 per tank. Each diet was administered into tanks to apparent satiation twice daily (8:00, 18:00). The food pellets were placed in a sealed bottle and the unused feed at the end of each feeding was weighed. Fish were reared and fed under 12-h light 12-h dark photoperiods. The feeding trial lasted for 71 days and each diet treatment was tested in triplicate.

### 4.3. Sample Collection and Measurement

At the end of the feeding trial, fish were fasted for 24 h. The body weight and body length of each fish in each tank were measured. Three fish in each tank were randomly selected and were euthanized using 100 mg/L tricaine methanesulfonate (MS-222, Sigma, St. Louis, MO, USA). Total body length (±0.1 mm) and body mass (±0.1 mg) were also measured before preservation using a digital scale and a pair of digital calipers for all fish and fish samples were dissected. Viscus and intestines mass (±0.1 mg) for each fish were removed and divided. Intestine samples for histology were fixed in 4% paraformaldehyde and the remaining sample was snap-frozen in liquid N_2_ and stored at −80 °C for later assay of biochemical indexes, microbial DNA, transcriptome analysis, and RNA quantification. 

### 4.4. Growth Performance and Intestine Histological Processing

Growth performance indices used for this study included: CF = (weight/(length)^3^) × 100; WG = ((W_f_ − W_i_)/W_i_) × 100; VSI = W_v_/W × 100; HSI = W_h_/W × 100 where W_i_ and W_f_ were the initial and final weights, respectively, Wv designates the viscerosomatic weight and W_h_ the hepatosomatic weight [43,64]. The results in growth performance were expressed as mean ± standard deviation (*n* = 90).

The intestines of experimental fish were collected and fixed in 4% paraformaldehyde (Servicebio Technology Co., Ltd., Wuhan, China) and then washed twice with PBS (Servicebio Technology Co., Ltd., Wuhan, China), then dehydrated in an ethanol/methanol series and embedded in paraffin. The paraffin tissue block was dyed with hematoxylin and eosin (HE). Histological images were photographed using an M8 Automatic Digital Slide Scanning System (Precipoint, Freising, Germany).

### 4.5. DNA Extraction, 16S rRNA Gene Sequencing and Data Processing

Microbial DNA was extracted from intestinal contents using the E.Z.N.A. Soil DNA Kit (Omega Bio-Tek, Norcross, GA, USA) according to the manufacturers protocol. The V4-V5 regions of the bacterial 16S ribosomal RNA genes were amplified using primers 515-F (5′-GTGCCAGCMGCCGCGG-3′) and 907-R (5′-CCGTCAATTCMTTTRAGTTT-3′) where the barcode was represented by an eight-base sequence unique to each sample (Illumina. San Diego, CA, USA). PCR reactions were performed in triplicate and amplicons were extracted from 2% agarose gels and purified using the AxyPrep DNA Gel Extraction Kit (Axygen Biosciences, Union City, CA, USA) according to the manufacturer’s instructions. Purified PCR products were quantified using a Qubit 3.0 instrument (Invitrogen, Carlsbad, CA, USA) and the pooled DNA product was used to construct an Illumina Pair-End library following Illumina’s genomic DNA library preparation procedure. The amplicon library was paired-end sequenced (2 × 250) on an Illumina MiSeq platform at Biozeron (Shanghai, China) according to the standard protocols. 

Sequences were clustered into OTUs at 100% similarity (identify) using the Deblur denoising algorithm that removed noise due to sequencing error [65]. Clusters of identical sequences allowed us to detect microbial changes at a fine scale resolution. The phylogenetic affiliation of each 16S rRNA gene sequence was analyzed by uclust algorithm (http://www.drive5.com/usearch/manual/uclust_algo.html, accessed on 3 March 2022) against the silva (SSU138.1) 16S rRNA database using a confidence threshold of 80% [66]. All statistical analysis was performed with the R stats package. Redundancy analysis (RDA) was employed to explore the relationship between environmental factors and bacterial communities. 

### 4.6. Sample Biochemical Parameters Assay 

The intestine samples were homogenized with ice-cold 0.9% saline solution (1 g: 9 mL) with a high-throughput tissue grinder (Shanghai Jingxin Industrial Development Co., Ltd., Shanghai, China). Processed samples were centrifuged at 4000 rpm at 4 °C for 10 min, and the supernatant was collected. Then, the supernatant was used to measure the intestine biochemical indexes including the activities of SOD, CAT, T-AOC, ACP, and AKP, and the content of MDA, GSH, and LDH. All biochemical indexes were processed with commercial kits (Nanjing Jiancheng Bioengineering Institute, Nanjing, China) and measured on a Multi-Mode Reader (Synergy HTX, Bio Tek, Santa Clara, CA, USA).

### 4.7. Intestine Transcriptome Analysis

#### 4.7.1. RNA Extraction, Library Construction and Sequencing

Total RNA was extracted with Trizol reagent (Invitrogen) according to the manufacturer’s protocol and then treated with RNase-free DNase I (Takara, Tokyo, Japan) to prevent DNA contamination. The concentration and quality of total RNA were determined by the Nanodrop 2000 (Thermo, Pittsburg, PA, USA) and the Agilent 2100 Bioanalyzer (Agilent, San Diego, CA, USA). The mRNA libraries of the 0 mg/kg (0–1, 0–2, 0–3), 0.2 mg/kg (0.2–1, 0.2–2, 0.2–3), 2 mg/kg (2–1, 2–2, 2–3) and 20 mg/kg (20–1, 20–2, 20–3) groups were created using a Truseq RNA sample prep kit (Illumina) according to the manufacturer’s protocol and were sequenced using an Illumina NovaSeq 6000 system.

#### 4.7.2. Quality Control and Assembly of the Transcriptome

Raw reads in FASTQ format were filtered with SeqPrep (https://github.com/jstjohn/SeqPrep, accessed on 30 November 2021) and were trimmed with Sickle (https://github.com/najoshi/sickle, accessed on 30 November 2021) to remove the adaptor sequences, low-quality sequences (quality value < 30) and reads with >10% poly-N. The Q20, Q30 and GC content of the clean data were then calculated. All subsequent analyses were based on the high-quality clean data and clean reads were mapped to the *C. carpio* genome (NCBI Acc. No. ASM1834038v1) using HISAT2 [67]. The mapped reads of each sample were obtained for the assembly of the transcriptome using StringTie (http://ccb.jhu.edu/software/stringtie/, accessed on 30 November 2021) [68].

#### 4.7.3. DEG Analysis and Functional Annotation

The fragments per kilobase of transcript per million mapped reads (FPKM) algorithm was used to normalize mRNA expression levels and DEGs between groups were defined using DESeq2 with the threshold of significance as the corrected *p* value < 0.05 and |log2 fold change| ≥ 1 [69]. Functional DEG annotations were performed via the KEGG pathway enrichment analysis [70]. The *p* value was calculated using Fisher’s test and was corrected using the Benjamini and Hochberg correction.

### 4.8. Analysis of Target Gene Expression

DEGs were randomly selected for verifying the mRNA expression profiling. Primer sequences for candidate genes were designed and synthesized based on C. carpio genome (NCBI Acc. No. ASM1834038v1). PCR primers are shown in Appendix A. The qRT-PCR methods were the same as that used in our previous studies [1]. In brief, qRT-PCR of DEGs were carried out on a CFX Connect Real-Time System (Bio-Rad, Hercules, CA, USA) using AG SYBR Green Premix Pro Taq HS qPCR Kit (Accurate Biotechnology, Hunan, China). Details of the qRT-PCR materials and program was set at 95 °C for 30 s, followed by 40 cycles at 95 °C for 5 s, 60 °C for 30 s. Each amplification reaction was run in triplicate. After finishing the program, the threshold cycle (CT) values were obtained from each sample. The relative gene expression levels were evaluated using the 2^−ΔΔct^ method. Specific primers were used to amplify the selected genes with β-actin as an internal standard gene. 

### 4.9. Statistical Analysis

All experiments were carried out at least in triplicate and all values are expressed as the mean ± S.E.M. Statistical analyses from this research were performed using SPSS statistical package version 26.0 (SPSS, Chicago, IL, USA) and OriginPro 2022 (www.OriginLab.com, accessed on 10 May 2022). Differences were determined by analysis of variance (ANOVA) and *p* < 0.05 was considered statistically significant.

## 5. Conclusions

Overall, the growth performance of *C. carpio* was significantly improved after 71 days of feeding with KM. KM not only improved the intestinal microbiota and enhanced intestinal nutrient absorption but also alleviated the oxidative stress state of the *C. carpio* intestines. KM induced cell transcription and proliferation processes through TGF-β1/Smad, MAPK, and Jak/Stat signaling pathways, thus significantly improving the growth performance of *C. carpio*. However, 20 mg/kg of KM may induce apoptosis of *C. carpio* intestinal cells, which is detrimental to nutrient absorption. KM exists at a concentration threshold of 0.2 to 2 mg/kg to control *C. carpio* intestinal cell proliferation or apoptosis. Based on our results, the recommended dose of KM supplementation applies to 0.2 mg/kg, and its use as a feed supplement has important implications for improving aquaculture. However, further research is needed to increase our comprehension of its mode of action.

## Figures and Tables

**Figure 1 ijms-23-11860-f001:**
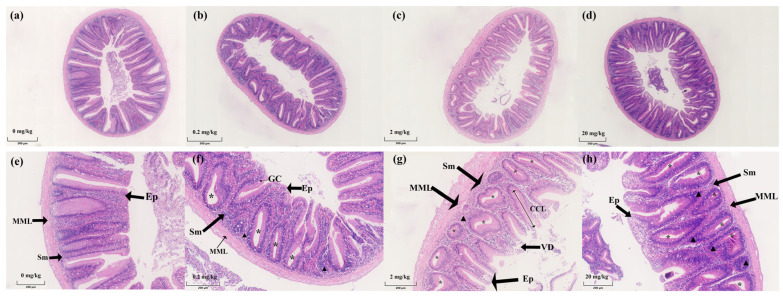
Transverse sections of the intestinal tract of carp chronically fed test diets containing KM: (**a**) 0 mg/kg group; (**b**) 0.2 mg/kg group; (**c**) 2 mg/kg group; (**d**) 20 mg/kg group (hematoxylin and eosin, HE, 5×); (**e**) 0 mg/kg group; (**f**) 0.2 mg/kg group; (**g**) 2 mg/kg group; (**h**) 20 mg/kg group (HE, 20×). Ep: epithelium, MML: muscular mucosa layer, Sm: submucosa, GC: goblet cells, triangular arrows: cell proliferation and disorder of arrangement, Asterisks: crypt cells.

**Figure 2 ijms-23-11860-f002:**
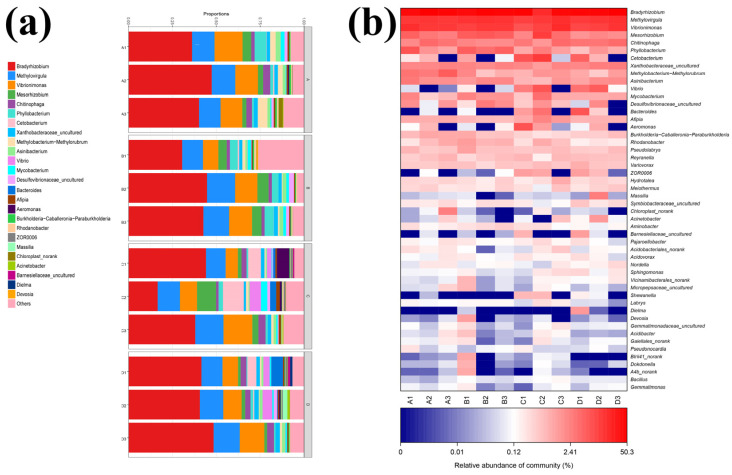
Relative abundance of the primary bacterial genera in the intestine of *C. carpio* grouped by KM treatment: (**a**) microbial community analysis (genus); (**b**) microbial community heatmap analysis (genus); sample A: 0 mg/kg; B: 0.2 mg/kg; C: 2 mg/kg; D: 20 mg/kg; each bar corresponds to an individual fish; bacterial classes showing a relative abundance <1% were designated as “others”; (**c**) the relative abundance of different flora in the intestine. Different superscript letters indicate significant differences for each pairwise comparison between treatments (*p* < 0.05).

**Figure 3 ijms-23-11860-f003:**
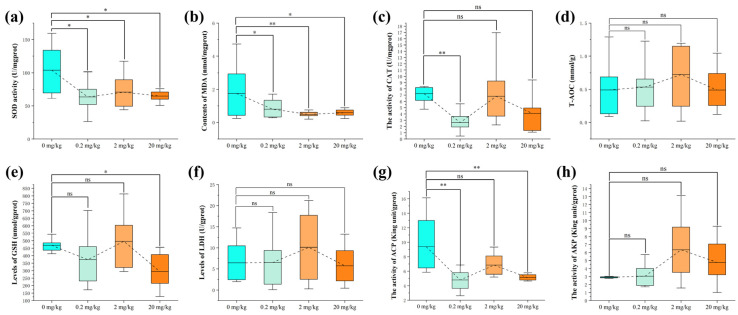
Changes in biochemical indices in the intestine of *C. carpio* fed by KM treatment: (**a**) SOD activities; (**b**) MDA levels; (**c**) CAT levels; (**d**) total antioxidant activity (T-AOC) levels; (**e**) GSH levels; (**f**) lactate dehydrogenase (LDH) levels; (**g**) alkaline phosphatase (ACP) activities; (**h**) AKP activities (data = mean ± standard error, *n* = 9). Asterisks denote that correlations were significant at *p* < 0.05 (*) and *p* < 0.01 (**) and “ns” denotes that correlations were no significant.

**Figure 4 ijms-23-11860-f004:**
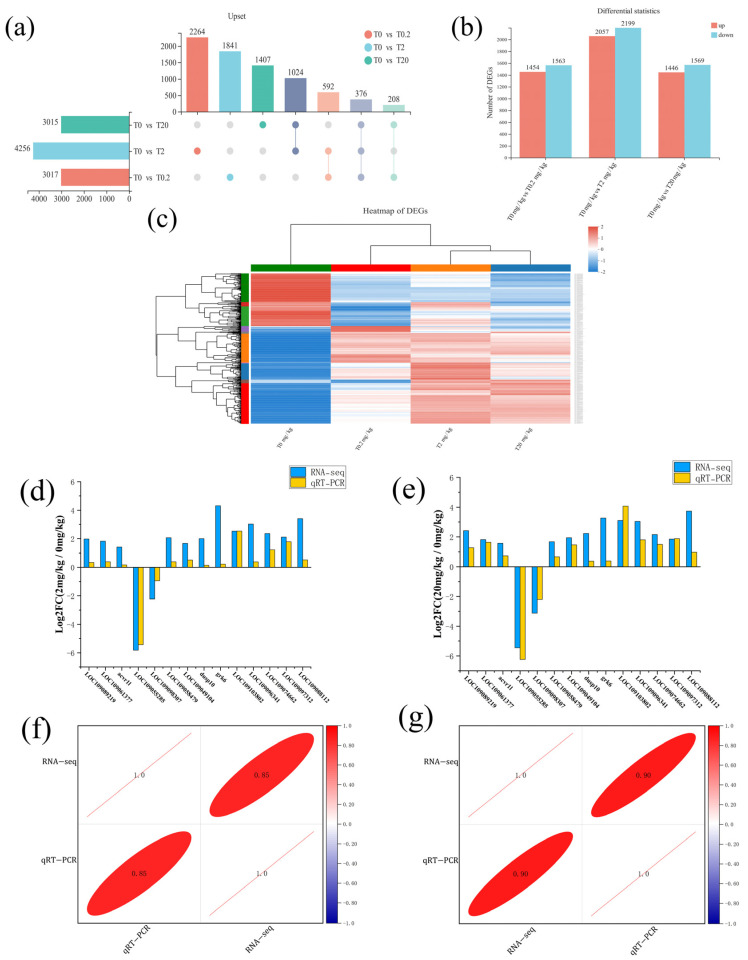
DEG Cluster profiling and validation of DEG data by qRT-PCR: (**a**) comparative results of transcripts between different groups: (**b**) DEG histogram after 71 days of feeding at the indicated concentrations of KM; (**c**) DEG clustering; (**d**) comparison of relative Log_2_^(fold change)^ between RNA-seq and quantitative (q) RT-PCR; 2 mg/kg group; (**e**) comparison of relative Log_2_^(fold change)^ between RNA-seq and qRT-PCR; 20 mg/kg group; (**f**) correlation analysis diagram of gene expression; 2 mg/kg group; (**g**) correlation analysis diagram of gene expression; 20 mg/kg group.

**Figure 5 ijms-23-11860-f005:**
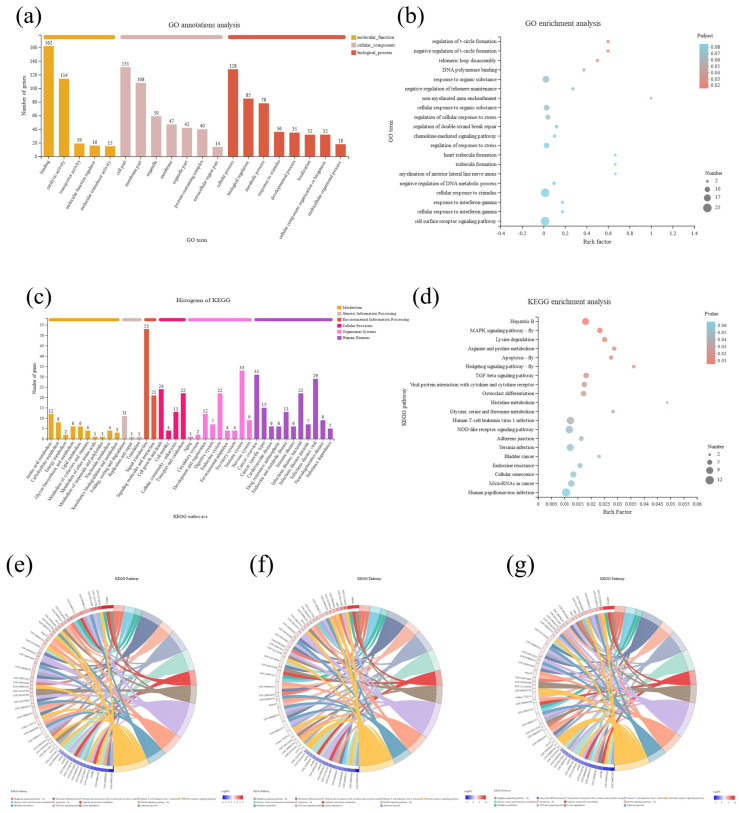
GO and KEGG analysis of DEGs from *C. carpio* fed different levels of KM: (**a**) GO annotation analysis of DEGs; (**b**) GO annotation enrichment analysis of DEGs; (**c**) KEGG analysis of DEGs; (**d**) KEGG enrichment analysis of DEGs; DEGs corresponding to the KEGG pathway in groups (**e**) 0.2, (**f**) 2, and (**g**) 20 mg/kg.

**Figure 6 ijms-23-11860-f006:**
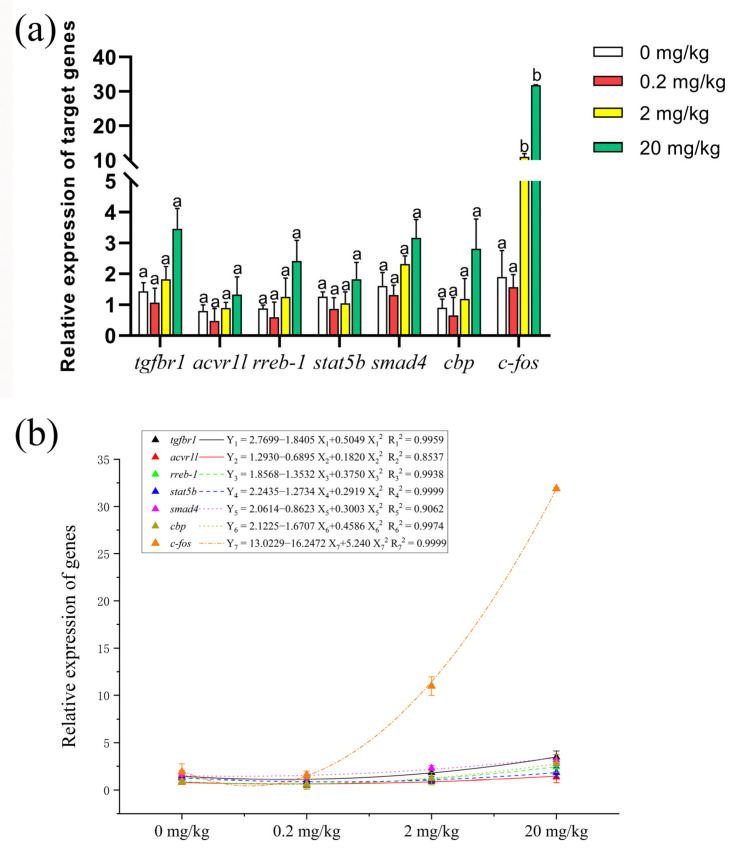
Relative expression of genes related to cellular processes in the *C. carpio* fed by different KM concentrations: (**a**) histogram of relative expression of target genes (data = mean ± standard error, *n* = 9); (**b**) comparative polynomial fit analysis of relative expression of genes. Bars of the same gene with different letters indicate a significant difference (*p* < 0.01, *n* = 9).

**Figure 7 ijms-23-11860-f007:**
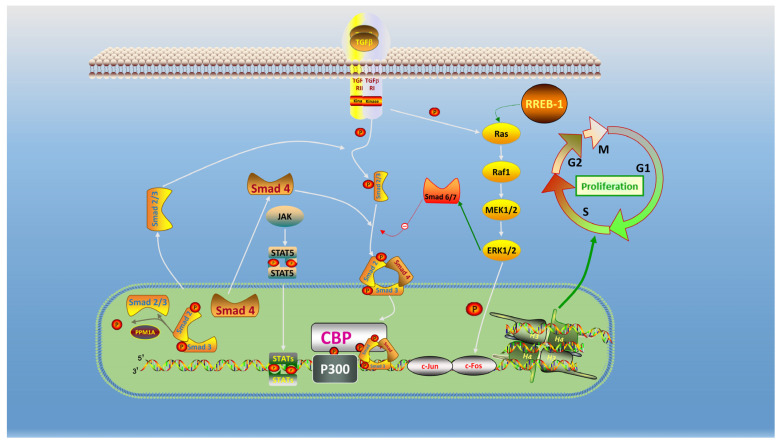
Relationships between TGF-β1/Smads, MAPK and Jak/Stat pathways involved in cell proliferation. Cell proliferation and apoptosis strictly regulate the homeostasis of intestinal cells. Signaling-coupled pathways that activate and promote cell proliferation and inhibit apoptosis act through TGF-β and RREB in response to stimulation by internal and external survival factors. TGF-β/Smad, ERK-MAPK activates CBP/p300 to promote cell transcriptional proliferation. Activation of the Jak-Stat pathway that phosphorylates STAT5 negatively regulates cellular transcription and prevents excessive cell proliferation.

**Table 1 ijms-23-11860-t001:** Growth performance of fish fed test diets for 71 days.

Treatment	Weight (g ± SD)	Length (cm ± SD)	CF	WG (%)	VSI (%)	HSI (%)
0 mg/kg	20.6 ± 1.2 ^a^	9.4 ± 1.1 ^ab^	2.4 ± 0.27 ^a^	2.92 ± 1.32 ^a^	0.07 ± 0.01 ^a^	0.02 ± 0.01 ^b^
0.2 mg/kg	25.9 ± 9.4 ^b^	9.9 ± 1.2 ^ab^	2.53 ± 0.19 ^b^	5.37 ± 2.31 ^b^	0.09 ± 0.02 ^b^	0.06 ± 0.02 ^a^
2 mg/kg	26.1 ± 9.9 ^b^	10.0 ± 1.1 ^a^	2.54 ± 0.2 ^b^	3.13 ± 1.56 ^a^	0.07 ± 0.01 ^ab^	0.04 ± 0.01 ^ab^
20 mg/kg	22.3 ± 7.5 ^ab^	9.4 ± 1.1 ^b^	2.64 ± 0.23 ^b^	3.27 ± 1.43 ^a^	0.07 ± 0.00 ^a^	0.03 ± 0.01 ^b^

Values are means ± SD of 90 replications in 4 groups. Different superscript letters (“a”, “b”, and “ab”) indicate significant differences for each pairwise comparison between treatments (*p* < 0.05).

## Data Availability

The authors declare that the data supporting the findings of this study are available within the article and its Appendix A.

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
