# Peer review of "Effects of Dietary Koumine on Growth Performance, Intestinal Morphology, Microbiota, and Intestinal Transcriptional Responses of Cyprinus carpio"

_ijms, 2022, doi:10.3390/ijms231911860_

Round 1
Reviewer 1 Report
In the paper “Effects of dietary koumine on growth performance, intestinal morphology, microbiota and intestinal transcriptional responses of Cyprinus carpio, authors have investigated aquatic immunostimulant and growth-promoting effects of Koumine alkaloid. The study has been carried out nicely, but the quality of presentation can be improved. The resolution of most of the figures is very poor. Increase the quality of figures for more clarity. Table 1, clearly indicate the superscript to show differences between groups. For example, “a” between 0 mg/kg and 0.2 mg/kg etc.
Reviewer 2 Report
The study is interesting but needs major revision.
1. Language needs edition as there are several informal writing in the text, for example (you must deeply read and revise the whole text):
2. Keywords: There should be no repeated words contained in the title.
3. Add the references in material and methods section.
4. The authors must use for relevant citations, for example:
- The effects of broccoli and caraway extracts on serum oxidative markers, testicular structure and function, and sperm quality before and after sperm cryopreservation
- The comparison of the effect of Origanum vulgare L. extract and vitamin C on the gentamycin-induced nephrotoxicity in rats
- Beneficial effects of Persian shallot (Allium hirtifolium) extract on growth performance, biochemical, immunological and antioxidant responses of rainbow trout Oncorhynchus mykiss fingerlings
- The use of dietary oak acorn extract to improve haematological parameters, mucosal and serum immunity, skin mucus bactericidal activity, and disease resistance in rainbow trout (Oncorhynchus mykiss)
- Effects of dietary vitamin C, thyme essential oil, and quercetin on the immunological and antioxidant status of common carp (Cyprinus carpio)
- The effect of dietary combined herbs extracts (oak acorn, coriander, and common mallow) on growth, digestive enzymes, antioxidant and immune response, and resistance against Aeromonas hydrophila infection in common carp, Cyprinus carpio
- The effects of combined inclusion of Malvae sylvestris, Origanum vulgare, and Allium hirtifolium boiss for common carp (Cyprinus carpio) diet: Growth performance, antioxidant defense, and immunological parameters.
5. Why these levels? How were they defined?
6. Revise references according to journal style (also in the text).
7. Please add the full name for the first occurrence of the abbreviation. Full text check.
8. When write fish name for the first time, add scientific name along with common name.
9. The conclusions are not informative. The main findings, limitations, and future work should be described in the manuscript.
10. The authors should update the list of references to contain recent publications in 2021 and 2022.
11. Figures: It is not completely clear.
Round 2
Reviewer 2 Report
Dear editor
The article is now acceptable in current form.